# What is current care for people with Long COVID in England? A qualitative interview study

David Sunkersing ![ORCID] ,[1] Mel Ramasawmy,[1] Nisreen A Alwan,[2,3] Donna Clutterbuck,[2] Yi Mu,[1] Kim Horstmanshof,[4] Amitava Banerjee ![ORCID] ,[1] Melissa Heightman[5]

[1]Institute of Health Informatics, University College London, London, UK
[2]Academic Unit of Primary Care and Population Sciences, Faculty of Medicine, University of Southampton, Southampton, UK
[3]NIHR Southampton Biomedical Research Centre, University Hospital Southampton NHS Foundation Trust, Southampton, UK
[4]University College London, London, UK
[5]University College London Hospitals NHS Foundation Trust, London, UK

**Correspondence to**
Dr David Sunkersing;
david.sunkersing@ucl.ac.uk

## ABSTRACT

**Objective** To investigate current care for people with Long COVID in England.
**Design** In-depth, semistructured interviews with people living with Long COVID and Long COVID healthcare professionals; data analysed using thematic analysis.
**Setting** National Health Service England post-COVID-19 services in six clinics from November 2022 to July 2023.
**Participants** 15 healthcare professionals and 21 people living with Long COVID currently attending or discharged (18 female; 3 male).
**Results** Health professionals and people with lived experience highlighted the multifaceted nature of Long COVID, including its varied symptoms, its impact on people's lives and the complexity involved in managing this condition. These impacts encompass physical, social, mental and environmental dimensions. People with Long COVID reported barriers in accessing primary care, as well as negative general practitioner consultations where they felt unheard or invalidated, though some positive interactions were also noted. Peer support or support systems proved highly valuable and beneficial for individuals, aiding their recovery and well-being. Post-COVID-19 services were viewed as spaces where overlooked voices found validation, offering more than medical expertise. Despite initial challenges, healthcare providers' increasing expertise in diagnosing and treating Long COVID has helped refine care approaches for this condition.
**Conclusion** Long COVID care in England is not uniform across all locations. Effective communication, specialised expertise and comprehensive support systems are crucial. A patient-centred approach considering the unique complexities of Long COVID, including physical, mental health, social and environmental aspects is needed. Sustained access to post-COVID-19 services is imperative, with success dependent on offering continuous rehabilitation beyond rapid recovery, acknowledging the condition's enduring impacts and complexities.

## STRENGTHS AND LIMITATIONS OF THIS STUDY

⇒ This study used in-depth, semistructured interviews to provide a rich account of current care, from participants across National Health Service England commissioned post-COVID-19 services in England.
⇒ Voluntary participation from individuals able to access a post-COVID-19 service could lead to an under-representation of certain respondent types.
⇒ As this study aimed to investigate perspectives from six post-COVID-19 sites, the findings may not be generalisable or representative of other post-COVID-19 sites in England.

## INTRODUCTION

Long COVID (or 'postacute sequelae of SARS-CoV-2 infection')[1,2] refers to a complex and poorly understood set of persistent symptoms that can continue for weeks, months or longer after an initial COVID-19 infection. First identified online from individuals with ongoing symptoms of COVID-19, it quickly gained recognition in formal clinical channels within months.[3] Estimates suggest at least 65 million people worldwide have Long COVID, with reported symptoms including extreme fatigue, breathlessness, cognitive impairments and depression.[1,4,5] For some, Long COVID leads to prolonged recovery, severely impacting their work, daily activities and social relationships.[6] Importantly, the recognition and understanding of Long COVID have been enhanced by the crucial role played by people with the condition themselves, including through supportive peer-to-peer interactions, where valuable networks of shared experiences and coping strategies have been exchanged.[7] However, as symptoms often present heterogeneously, diagnosis and treatment are challenging.[1,4,8]

In September 2020, recognising the distinctive and persistent healthcare needs of individuals with Long COVID,[3,9] the Royal College of General Practitioners called for the establishment of a national network of 'post-COVID-19' services. In response, National Health Service England (NHSE) committed to rapidly expanding new and strengthened rehabilitation centres.[10] This occurred when only a minority of Clinical Commissioning Groups had successfully implemented post-COVID-19 services, reflecting challenges in

adapting healthcare services for people with Long COVID. In October 2020, NHSE implemented a comprehensive five-part package to bolster support for people with Long COVID.[11] This included the release of new guidance on managing Long COVID, the introduction of an online rehab service ('Your Covid Recovery'), designated Long COVID clinics/post-COVID-19 services (to treat Long COVID symptoms, deliver holistic assessments with a multidisciplinary team, offer medical support, diagnostics and referral and rehabilitation where needed),[12] NIHR-funded research and establishing a new NHSE Long COVID task force.[12 13] These initiatives underscored the evolving understanding of Long COVID and the imperative for healthcare services to adapt rapidly to effectively support those navigating the complex and prolonged effects of the virus.

For people with suspected Long COVID, UK guidance advises beginning with a general practitioner (GP) consultation for an initial assessment.[14] This involves discussing symptoms, ruling out underlying conditions and evaluating their impact on daily life. Referral to specialised services like rehabilitation or post-COVID-19 services may follow if symptoms severely disrupt daily functioning, in accordance with NICE and WHO recommendations.[15 16] Despite this guidance, care for people with Long COVID can vary—and people have reported difficulties in accessing care,[17 18] as well as varied experiences in their treatment and support.[19] For healthcare professionals, the existing complexity in addressing Long COVID care is compounded by ongoing staff shortages, under-resourcing and insufficient funding.[20–22]

Currently, there is lack of in-depth, critical analyses concerning patient experiences in their ongoing care for Long COVID.[17 23 24] Similarly, the challenges of implementing effective post-COVID-19 services contribute to the limited evidence on healthcare professionals' perspectives in experiences of managing Long COVID care.[18 25] To effectively manage and improve care for people living with Long COVID, it is crucial to evaluate current healthcare approaches for this specific patient group and their healthcare professionals. Our study aimed to address these gaps by conducting an in-depth investigation into current care within a sample of newly established post-COVID-19 services in England. This paper presents findings from semistructured interviews conducted with both individuals experiencing Long COVID and healthcare professionals providing Long COVID care.

## METHODS
### Study design and setting
People living with Long COVID, either discharged or under follow-up from six sites across England, were invited to participate virtually over Microsoft Teams in either semistructured interviews or focus groups (from November 2022 to July 2023). Healthcare professionals providing Long COVID care at each of the six sites were also invited to participate.

In addition to the above, the study inclusion criteria were >18 years old, English speaker and competent to provide written consent.

### Recruitment of people with Long COVID
The service manager and clinical lead for each of the six post-COVID-19 services, which were identified as part of a previous study, known as Symptoms, Trajectory, Inequalities and Management: Understanding Long-COVID to Address and Transform Existing Integrated Care Pathways,[26] were asked to review their database of people living with Long COVID, including those that may have signed up to take part in the research. Depending on the most appropriate site-specific means to approach people living with Long COVID, we contacted (via email, via an online Long COVID group at one site (led by the post-COVID-19 service) and/or via an internal hospital communication app at another site (MyCare application)) them to advertise the project and request their involvement. Voluntary response sampling (ie, where participants had volunteered themselves to participate) was employed in this study to accommodate the unique challenges and vulnerabilities of people with Long COVID—and, importantly, recognising that those affected may experience limitations in time and energy. Although this sampling strategy has limitations, we felt that this sampling strategy offered a more feasible and considerate approach for gathering valuable insights into the experiences of people with Long COVID.

Regular contact with all six post-COVID-19 services was made throughout the data collection period to maximise participation. The principle of saturation guided the sample size, and semistructured interviews were conducted until additional participants did not yield new information.

### Recruitment of healthcare professionals
Apart from one site, each service manager and clinical lead additionally identified Long COVID healthcare professionals who were willing to participate in the study. We chose this approach—voluntary response sampling—as we considered it the most pragmatic and efficient method for collecting data.

In our healthcare professional interviews, we also investigated the route of individuals with Long COVID through a site's post-COVID-19 services to understand the structure, delivery and accessibility of care of the Long COVID pathway.

### Data collection and semistructured interview
Each participant was asked to describe their experience of care, including referral, assessment, diagnosis and treatment (see online supplemental materials 1 and 2 for full interview guides). For people living with Long COVID, the aim of the semistructured interviews/focus groups was to gain a greater understanding of their perception of care including any barriers to accessing care, their experience of the care pathway and any outcomes they experienced

due to treatment. For healthcare professionals providing care for people living with Long COVID, we wanted to understand the delivery of Long COVID care from their perspective, including any challenges, historical or current. Semistructured interviews/focus groups were selected as the data collection approach because they can provide comprehensive, profound and detailed narratives of participants' experiences, viewpoints and thoughts. They also strike a balance between flexibility and structure, allowing for an in-depth exploration of individual experiences with Long COVID while providing a framework for key topics. This approach enabled us to adapt the interview based on participants' responses, fostering a richer understanding of their perspectives and capturing nuanced information that may not have emerged in more rigidly structured interview formats. The semistructured interviews were conducted by DS and MR via Microsoft Teams, who are both experienced qualitative researchers. Semistructured interviews were conducted with both people with Long COVID and healthcare professionals until saturation was achieved and no new information was forthcoming.

## Data analysis

All participants in this study opted to take part in a semistructured interview—that is, no focus groups—between November 2022 and July 2023. All semistructured interviews were (with participant consent) audio recorded and automatically transcribed verbatim in Microsoft Teams using the 'transcribe feature', which converts speech to a text transcript with each speaker individually separated.

An inductive, data-driven approach[27] to data analysis was undertaken by DS and MR. First, each transcript was reviewed for accuracy and initial codes were generated using support from the qualitative analysis software programmes ATLAS.ti (V.5.8.0)[28] and NVivo V.12.[29]

Following initial code generation, these were reviewed and further refined. The process was concluded when an agreement that the overarching themes effectively encapsulated the narratives present in the data for both people with Long COVID and healthcare professionals.

Three overarching themes were generated ((1): understanding Long COVID: symptoms and lived realities, (2) Long COVID care: non-specialist service experiences and (3) Long COVID care: post-COVID-19 service experiences) and a number of supporting, subsidiary subthemes.

Two researchers (DS and MR) met regularly to discuss the transcripts and discuss whether the overarching themes were reflective of the presented data. These findings were also discussed with AB (and patient representatives) for review. After review, there was agreement that the themes reflected the transcripts appropriately.

This approach of generating shared overarching themes for both people with Long COVID and healthcare professional perspectives was done to enable a deeper understanding of shared and divergent perspectives—an approach used in other studies.[30 31]

## Patient and public involvement

A patient and public involvement (PPI) group (n=11) features in our regular management and oversight meetings. This group comprises individuals who either have Long COVID or provide care for someone with Long COVID. PPI members have contributed to the research components of this study, including informing the semistructured interview design and approach. We also discussed the findings of this study with our PPI to facilitate our interpretations of the data. The PPI group will continue to be involved in the dissemination of findings.

## RESULTS

A total of 21 people with Long COVID were interviewed from 4 of the 6 possible sites (10 from service a, 8 from service b, 2 from service c, 0 from service d, 1 from service e and 0 from service f). For people with Long COVID, the nature of their symptoms, privacy concerns, emotional challenges and/or ongoing medical commitments may have impacted participation.

All six post-COVID-19 services were approached and contacted, with five services able to identify healthcare professionals willing to participate in the study. A total of 15 healthcare professionals were interviewed (5 from service a, 4 from service b, 3 from service c, 0 from service d, 1 from service e and 2 from service f).

We believe the smaller than anticipated sample size in this study can be attributed to staff shortages, existing pressures within the NHS and time availability. At one specific service, the absence of semistructured interviews with healthcare professionals can be attributed to unforeseen circumstances, which hindered the engagement and participation of relevant personnel in the study.

Characteristics of people living with Long COVID are detailed in table 1 and characteristics of healthcare professionals providing Long COVID care are detailed in table 2.

We found that a generic Long COVID pathway was followed by many sites (figure 1), although comprehensiveness of testing and/or referral ability varied, based on available staffing, other resources and local infrastructure.

From our healthcare professional interviews, we were able to establish the service model used at each post-COVID-19 service:
1. Face to face.
2. Hybrid (predominantly virtual; some aspects face to -face).
3. Face to face.
4. Hybrid (dependent on patient).
5. Hybrid (dependent on patient).
6. Hybrid (dependent on patient—home visits, if necessary).

Information from post-COVID-19 service '4' was obtained from publicly available data, as no direct interview was conducted with this site.

**Table 1** Characteristics of people living with Long COVID

| Characteristic | | Frequency | % |
|---|---|---|---|
| Sex | Male | 3 | 14 |
| | Female | 18 | 86 |
| Age | 18–27 | 0 | 0 |
| | 28–37 | 1 | 5 |
| | 38–47 | 12 | 57 |
| | 48–57 | 6 | 29 |
| | 58–67 | 0 | 0 |
| | 68–77 | 2 | 10 |
| Ethnic group | White British | 12 | 57 |
| | White—any other white background | 3 | 14 |
| | Asian or Asian British—Indian | 3 | 14 |
| | Other ethnic groups—any other ethnic group | 2 | 10 |
| | Not stated | 1 | 5 |
| Languages spoken | English (only) | 14 | 67 |
| | Multiple (Including English) | 7 | 33 |
| Highest educational qualification | General Certificate of Secondary Education (GCSE)/ equivalent | 1 | 5 |
| | A-level | 1 | 5 |
| | Apprenticeship | 1 | 5 |
| | Undergraduate degree | 8 | 38 |
| | Master's degree | 8 | 38 |
| | PhD | 1 | 5 |
| | Not stated | 1 | 5 |
| Profession (current or prior to COVID-19/Long COVID (LC)) | Creative arts or design | 2 | 10 |
| | Education | 1 | 5 |
| | Engineering | 1 | 5 |
| | Financial | 2 | 10 |
| | Healthcare (clinical facing role) | 9 | 43 |
| | Marketing, advertising or PR | 1 | 5 |
| | Public services or administration | 2 | 10 |
| | Unemployed | 1 | 5 |
| | Retired | 1 | 5 |
| | Not stated | 1 | 5 |
| Comorbidities | Yes | 16 | 76 |
| | No | 4 | 19 |
| | Not stated | 1 | 5 |
| LC onset | January–April 2020 | 6 | 29 |
| | May–August 2020 | 2 | 10 |
| | September–December 2020 | 2 | 10 |
| | January–April 2021 | 1 | 5 |
| | May–August 2021 | 4 | 19 |
| | September–December 2021 | 0 | 0 |
| | January–April 2022 | 5 | 24 |
| | May–August 2022 | 1 | 5 |

Continued

**Table 1** Continued

| Characteristic | | Frequency | % |
|---|---|---|---|
| | September–December 2022 | 0 | 0 |
| Post-COVID-19 service attendance | January–April 2020 | 0 | 0 |
| | May–August 2020 | 2 | 10 |
| | September–December 2020 | 3 | 14 |
| | January–April 2021 | 3 | 14 |
| | May–August 2021 | 0 | 0 |
| | September–December 2021 | 2 | 10 |
| | January–April 2022 | 1 | 5 |
| | May–August 2022 | 4 | 19 |
| | September–December 2022 | 5 | 24 |
| | January–April 2023 | 1 | 5 |

Three overarching themes were identified and are elaborated further with example supporting quotations in tables 3–5. We have refrained from attributing quotes to specific post-COVID-19 services to avoid potential identification of specific services and participants.

### Theme 1: Understanding Long COVID: symptoms and lived realities

Within this theme, people living with Long COVID reported on several factors relating to their Long COVID experiences. The 'physical' subtheme underscores the persistent battles with fluctuating symptoms, highlighting the complexities that impact daily life (quotes 1–2, table 3). The 'mental health' subtheme captures the emotional toll of uncertainty, anxiety and stress related to the nature of the condition (quotes 3–4, table 3). The 'social' subtheme explores 'loss of identity or self', as people living with Long COVID grapple with both understanding and support (quote 5, table 3). Lastly, the theme pays heed to the often-overlooked environmental factors, examining how the interplay between personal surroundings and the broader context can shape the Long COVID journey (quote 6, table 3). The interviews additionally established an evident impact of Long COVID on their working environment—and, for some, a lack of universal appreciation and/or adjustment from colleagues (quotes 7–8, table 3).

Overall, these subthemes weave a narrative that underscores the complex challenges faced by people living with Long COVID.

Healthcare professionals, in their semistructured interviews, echoed and acknowledged the multifaceted nature of Long COVID, aligning with the reports and experiences shared by people living with Long COVID in this study (quotes 9–10, table 3).

By addressing physical, mental, social and environmental dimensions, this theme highlights the holistic nature of care needed for people living with Long COVID navigating the complexities of the condition and underscores the importance of a comprehensive and integrated approach to their well-being.

### Theme 2: Long COVID care: non-specialist service experiences

This theme reflects the intricate journey that people living with Long COVID undertake within the healthcare system in non-specialist settings (ie, outside of a post-COVID-19 service). This theme encompasses a participant's experiences, challenges and interactions within the healthcare system, highlighting their encounters with medical consultations and also support outside of healthcare settings.

Our study found that securing an initial appointment with a GP was challenging, both in terms of contact (quote 1, table 4) but also with respect to waiting times. Several people with Long COVID described lengthy wait times for appointments and a lack of efficient continuity in care (quotes 2–3, table 4). Moreover, our study reports encounters from people living with Long COVID where, after securing an appointment, their GP seemed to avoid comprehensive discussions regarding the intricacies of their Long COVID and/or people reported frustrations with the service received (quotes 4–5, table 4). Some emphasised a sense of being rushed or their concerns being sidelined, highlighting the significance of effective communication between people living with Long COVID and healthcare professionals (quote 6, table 4). These can all further exacerbate the challenges of Long COVID management.

Despite these comments, however, some people living with Long COVID were keen to make a note of the positive aspects that their GP had brought about (quotes 7–8, table 4). However, many people with Long COVID were especially keen to highlight the profound impact of community support networks (quotes 9–10, table 4). They emphasised the reassurance and validation found in shared experiences within online communities, as well as the crucial role played by friends and family in offering understanding and solidarity during

**Table 2** Characteristics of healthcare professionals providing Long COVID care

| Characteristic | | Frequency | % |
|---|---|---|---|
| Sex | Male | 3 | 20 |
| | Female | 12 | 80 |
| Age | 18–27 | 1 | 7 |
| | 28–37 | 5 | 33 |
| | 38–47 | 3 | 20 |
| | 48–57 | 2 | 13 |
| | 58–67 | 0 | 0 |
| | 68–77 | 0 | 0 |
| | Not stated | 4 | 27 |
| Ethnic group | White British | 6 | 40 |
| | White—any other white background | 2 | 13 |
| | Asian or Asian British—Indian | 2 | 13 |
| | Other ethnic groups—any other ethnic group | 2 | 13 |
| | Not stated | 3 | 20 |
| Profession | Clinical psychologist | 1 | 7 |
| | Clinical trials manager | 1 | 7 |
| | Doctor (gastro) | 1 | 7 |
| | Exercise physiologist | 2 | 13 |
| | Lead therapist (physio) | 1 | 7 |
| | Nurse | 1 | 7 |
| | Nurse (service lead) | 1 | 7 |
| | Occupational health physician and post-COVID-19 service (medical lead) | 1 | 7 |
| | Physiotherapist | 1 | 7 |
| | Post-COVID-19 lead | 1 | 7 |
| | Rehabilitation medicine consultant | 1 | 7 |
| | Respiratory physiotherapist | 1 | 7 |
| | Senior specialist physiotherapist | 1 | 7 |
| | Team lead post-COVID-19 Service | 1 | 7 |
| Have they had Long COVID? | Yes | 3 | 20 |
| | No | 8 | 53 |
| | Not stated | 4 | 27 |

To maintain participant anonymity, it is not possible to give further details on their roles.

the challenging journey of coping with long-lasting symptoms.

Lastly, we spoke to people with Long COVID who, though receiving some form of care outside of a post-COVID-19 service, were told that it was limited in addressing all aspects of their Long COVID (quote 11, table 4). While their experiences with these healthcare professionals were positive, the need for dedicated resources and expertise to effectively manage the intricate aspects of Long COVID was still apparent.

This theme particularly underscores the importance of fostering an environment where people living with Long COVID can (a) contact their healthcare provider and (b) openly express their experiences without fear of disbelief.

Importantly, the theme sheds light on the complexities and 'patient–provider' dynamics of Long COVID care, emphasising the need for improved 'patient–provider' communication, support and more efficient pathways within the healthcare system. Finally, the significance of a robust support network, where an individual's experiences feel validated cannot be overstated in the context of Long COVID.

### Theme 3: Long COVID care: post-COVID-19 service experiences

As noted in theme 2, participants recount their initial struggles in accessing appropriate care, emphasising the importance and need for specialised post-COVID-19

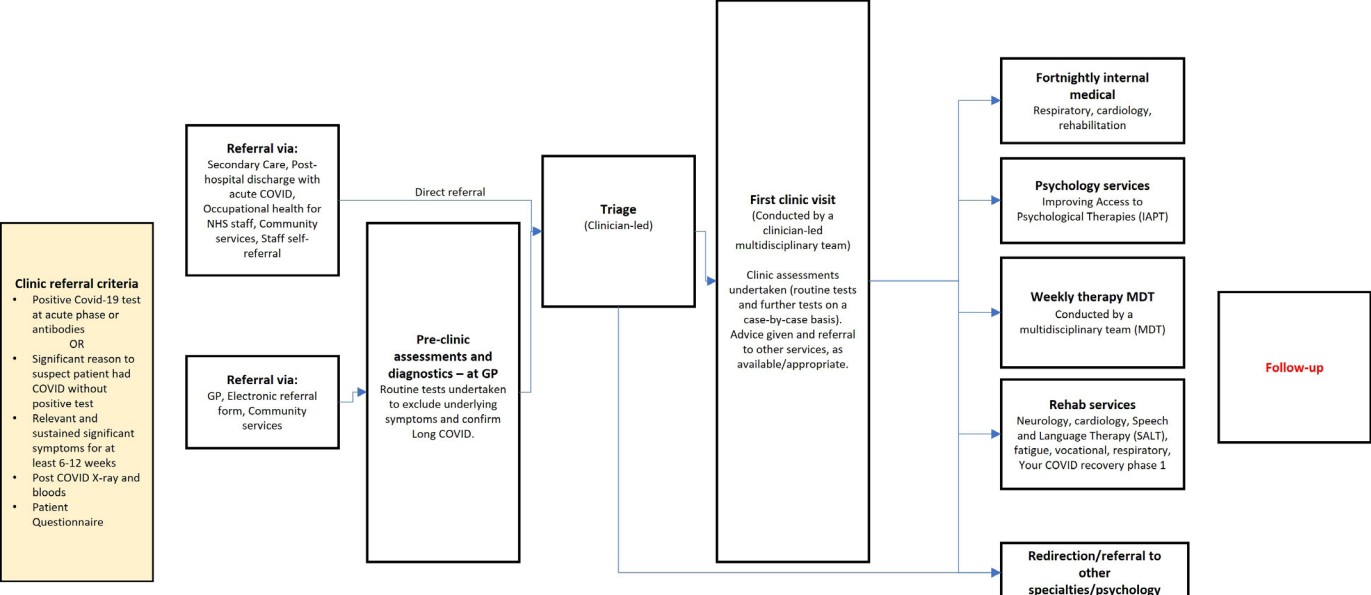

**Figure 1** Long COVID patient pathway. GP, general practitioner; NHS, National Health Service.

services. The establishment of dedicated services tailored to people living with Long COVID has also offered a sense of validation and recognition (quotes 1–2, table 5). For many people with Long COVID, these services represent more than just medical facilities; they serve as havens where individuals, perhaps previously unheard, find their voices listened to. Beyond medical reassurance, the services provide a space where the legitimacy of their symptoms and experiences are acknowledged, fostering a sense of validation and understanding that transcends merely physical symptoms. However, some people with

Long COVID suggested that they expected more comprehensive or hands-on support from these post-COVID-19 services (quote 3, table 5). Nevertheless, these services signify a commitment to understanding the complexities of Long COVID and acknowledging its lasting effects. Overall, people living with Long COVID appear to appreciate this understanding and Long COVID expertise.

The semistructured interviews with healthcare providers shed light on the experiences they faced while operating a Long COVID Service. Across all sites, the healthcare professionals confronted a multitude of obstacles, with a

| Table 3 | Theme 1: Understanding Long COVID: symptoms and lived realities |
|---|---|
| **Subtheme** | **Example supporting quote** |
| Physical | 'I've got symptoms of that I've never experienced in a life before, you know, the aching joints, the headaches.' (Participant T, Male, Age Range 68–77). |
| Physical | '…the first few days, it was a lot of kind of like coughing, breathlessness, fevers, et cetera. And then kind of over the first few days starting to get kind of extreme tiredness and fatigue…' (Participant S, Female, Age Range 28–37). |
| Mental health | '…there's a lot going on from a kind of stress, emotional sort of perspective.' (Participant J, Female, Age Range 38–47). |
| Mental health | '…the most thing from [Long] Covid that affected us is our mental health and the lack of sleep.' (Participant H, Female, Age Range 48–57). |
| Social | 'You know I'm shadow of the person I was and it's very hard to to accept that…' (Participant C, Female, Age Range 38–47). |
| Environmental | 'So instead of going into a cafe, for example, we'll get a coffee and sit in the park because there's less risk. So I'm still trying to maintain a life, but it's completely different.' (Participant N, Female, Age Range 38–47). |
| Working environment | '…and actually working online, I think is one of the really toughest things if you've got Long COVID.' (Participant A, Female, Age Range 48–57). |
| Working environment | 'You know they don't even give me time off for my appointments… sometimes I have to do them at work if I like. Take phone calls at work. You know it's really difficult.' (Participant E, Female, Age Range 38–47). |
| Multidimensional | 'Most of the patients come with cluster of symptoms, majority fatigue, breathlessness and also muscle aches, joint pains and the brain fog is also very common. The same cognitive issues forgetfulness and some with cardiac.' (Nurse (Service Lead)). |
| Multidimensional | '…and physical symptoms with long COVID patients and they are often burdened with quite a lot of mental health issues just because of the nature of the illness.' (Doctor (Gastro)). |

**Table 4** Theme 2: Long COVID care: non-specialist service experiences

| Subtheme | Example supporting quote |
|---|---|
| Perceived Lack of Support | [Regarding their GP Practice] 'It's impossible. You can't contact them in any way by phone or on the portal, and so you send them emails, and then no one, no one replies. They just ignore you.' (Participant E, Female Age Range 38–47). |
| Waiting times | 'Yeah, I think that's the like having to wait to 12 weeks kind of variability on the GP's.' (Participant S, Female, Age Range 28–37). |
| Waiting times | 'To see GP it was nightmare. I don't know if it's something to do particularly with my particular surgery…In terms of Long COVID clinic, it was in on average between three and six months.' (Participant F, Female, Age Range 48–57). |
| Frustrations with primary care service | '… you get into the into the, into the the GP's office…and then you get 5 minutes. That's all really 5, possibly less than 10, less than 10 anyway. A lot less than 10. And they're under pressure…it's it's not good.' (Participant T, Male, Age Range 68–77). |
| Frustrations with primary care service | 'We spoke to our practice manager—they didn't even process my husband's kidney scans. They didn't even tell him that he needed it from his CT—that there's something wrong with my husband's kidneys. So you know, we've been treated so so badly. But we can't move GPs again…they're all just as bad as each other, and moving won't help us.' (Participant H, Female, Age Range 48–57). |
| Negative clinical engagements | '…I was very much dismissed by the GP—she was quite unkind, actually. I think she just thought I was a bit nuts. I think she just thought its kind of PTSD, gone a bit nuts and very dismissive…' (Participant J, Female, Age Range 38–47). |
| Positive clinical engagements | '[I] was under so many people, I couldn't remember what's going on. She [the GP] had a list and she would just say, right, what's happening with this one? What's happening with that? Let's keep a track of everything. And it was really good because someone was keeping an eye out that I wasn't missing out anything or something need chasing.' (Participant E, Female, Age Range 38–47). |
| Positive clinical engagements | 'And they've [regarding their clinicians] been more than amazing like I can't even tell you. They they've they've listened to me. They've cared for me when they were really worried. I had some problems with work and sick leave, and whatever my GP was like, I'm gonna call you in two weeks' time just to see how you are, you know, unnecessarily, really, but just kindness…I can't praise them highly enough.' (Participant E, Female, Age Range 38–47). |
| Support networks | 'So the Facebook community…that community was was amazing because you knew for the first time that it wasn't just you and it wasn't. You weren't alone. So that was incredible.' (Participant N, Female, Age Range 38–47). |
| Support networks | '… there's sort of support from friends also with Long COVID or you know, and obviously friends and family members.' (Participant G, Female, Age Range 38–47). |
| Clinical Challenges | 'The respiratory physios…were, you know, still being very nice and still being very supportive but basically saying, 'look, we we feel we've got to the end of what we can do. We're not Long COVID specialists, we're not funded for for doing Long COVID we think that any residual problems in your breathing patterns are all mixed up with autonomic dysfunction.' (Participant G, Female, Age Range 38–47). |

significant portion stemming from the pervasive clinical uncertainty surrounding Long COVID and the absence of standardised training programmes. As a result, many healthcare providers had to rely on self-directed learning to stay updated and effective in their roles (quote 4, table 5). Furthermore, the persistent staffing and capacity issues within these specialised services meant healthcare teams often could not deliver the best possible care for people dealing with the long-lasting effects of COVID-19 (quotes 5–6, table 5). The semistructured interviews additionally highlighted the clinical uncertainty surrounding the best management practices for Long COVID (quote 7, table 5), indicating a pressing need for further research and guidance (eg, by having models of care commissioned to achieve this) in this emerging field. Healthcare professionals also underscored the difficulties of navigating integrated care systems (quote 8, table 5), emphasising the need for improved collaboration and coordination among various healthcare entities to better address the multifaceted needs of people with Long COVID.

Despite these challenges, for many post-COVID-19 services, while initially there were substantial clinical unknowns, there was now a sense of greater clinical expertise and confidence, resulting in more refined approaches in Long COVID care (quote 9, table 5).

## DISCUSSION

In our study, which examined perspectives from people with lived experience of Long COVID and healthcare professionals, we found three overarching themes: (1) understanding Long COVID: symptoms and lived realities, (2) Long COVID care: non-specialist service experiences and (3) Long COVID care: post-COVID-19 service experiences.

Our first theme resonates with contemporary research that underscores the intricate, individual and varied nature of the Long COVID journey. Our findings from people with Long COVID align with studies that have emphasised the diverse range of symptoms of this condition,[32] highlighting the importance of personalised care models that account for the multifaceted dimensions of Long COVID. The emphasis from people living with Long COVID on mental health aligns with other research,[33]

**Table 5** Theme 3: Long COVID care: post-COVID-19 service experiences

| Subtheme | Example supporting quote |
|---|---|
| Post-COVID-19 service experiences | 'They [post-COVID service] made me feel heard. But it wasn't until they got involved that I got all the tests [referring to medical tests and referrals]…But also, they gave me a community through their Facebook.' (Participant L, Female, Age Range 48–57). |
| Post-COVID-19 service experiences | [About the post-COVID service] 'It's reassurance as well. And then if they find something they obviously can tell you straight away. You don't need to go and look for for something, and then it's just even acknowledgement of of somebody understand you. And and know what what is going in your body and you are not on your own…so that's very important.' (Participant F, Female, Age Range 48–57). |
| Post-COVID-19 service experiences | 'Yeah, I feel like I was a bit disappointed with my like long COVID clinic. It was just like a 60 minute phone call.' (Participant S, Female, Age Range 28–37). |
| Post-COVID-19 service experiences (healthcare professional (HCP) perspective) | 'I think everybody's learning from each other, but no, not not haven't really kind of accessed like a formal training day. Like I've dipped into webinars and bits of things that I've seen that people are colleagues. I know that are doing that I'm kind of interested in.' (Physiotherapist). |
| Post-COVID-19 service experiences (HCP perspective) | 'The psychology offer could maybe be a bit stronger. I mean, it's quite good that we even consider it and we have psychologists in, but we just haven't been able to fill the role and then issues with space and staffing. Again, it's just something that's fallen to the wayside.' (Doctor (Gastro)). |
| Post-COVID-19 service experiences (HCP perspective) | 'They didn't have the capacity to take on any of this work. So we we wanted to have very much a multispecialty approach to managing long COVID. But our respiratory physicians and cardiologists didn't have the time available to to join the team. So the challenges have been that from a medical perspective, it's very much fallen solely on me.' (Rehabilitation Medicine Consultant). |
| Post-COVID-19 service experiences (HCP perspective) | '…working with people with long COVID can feel quite daunting unless you've seen a lot of it. And the confidence that comes with kind of managing that clinical uncertainty of what's in front of you only comes through with experience.' (Physiotherapist). |
| Post-COVID-19 service experiences (HCP perspective) | '…a key barrier…is how we work across the integrated care system. So we're all on different electronic records and the information governance agreements extend to some things and not others. We don't yet have sight of how our patients are doing. You know, we assess them in clinic and we'll send them to rehab, but we don't really get hearing about their outcomes and that that is that is problematic both in terms of how we assess whether what we're doing is effective and and how we coordinate care, but it also limits us in our delivery.' (Physiotherapist). |
| Post-COVID-19 service experiences (HCP perspective) | 'Diagnostics wise, I think at the beginning we did a lot of tests because we really didn't know what was going on. And as we've grown our clinical confidence and expertise…the number of tests reduced…' (Post COVID Lead). |

which showed the psychological and sometimes traumatic toll of living with Long COVID, further stressing the need for integrated mental health support as part of comprehensive Long COVID management strategies.

Our study additionally highlighted that the impact of Long COVID on people can extend well beyond its physiological manifestations, significantly affecting their immediate surroundings and (if applicable), their working environments. This study's findings corroborate with existing literature,[34 35] underlining the interplay between people living with Long COVID and the broader contexts in which they reside and work. The disruptions observed in social circles and routines align with the observations made in other studies,[36 37] highlighting the persistent challenges people living with Long COVID face as they navigate the unpredictable nature of Long COVID symptoms. Our study additionally echoes the experiences of those with cognitive impairments and reduced energy levels from Long COVID, resonating with other findings[38 39] that have documented the diverse impacts of Long COVID on professional functioning and adjustments required in work environments.[38 39] Notably, our findings indicated that the uncertainty caused by the lack

of an effective treatment along with debilitating symptoms and the potential disruption across various aspects of daily life, makes living with long COVID especially challenging.

The healthcare professional's perspectives of symptoms resonated with those of the patients, highlighting a shared understanding of the complex nature of Long COVID—and the multifaceted impact it can cause. Overall, these two perspectives underline the urgent need for supportive structures that acknowledge the multifaceted dimensions of Long COVID, both within the personal and professional spheres, ensuring comprehensive and responsive care.

Our second theme, which exclusively examined the perspectives of people with Long COVID, underscored their frustrations with the accessibility and responsiveness of their GP practice and described challenges in contacting them and/or receiving a response. Prolonged waiting times were frequently reported in our semistructured interviews, exacerbating their difficulties in accessing care. These experiences align with the literature[40 41] and highlight the need for improved access to healthcare services for Long COVID patients and

adequate support structures for managing this complex condition within primary care settings.

The experiences shared by people living with Long COVID in our study highlight a disconcerting circumstance wherein some GPs and other healthcare professionals seemingly avoid engaging in thorough discussions about the complexities of Long COVID. The people living with Long COVID consistently express feelings of being hurried through appointments or having their concerns dismissed. These findings align with emerging observations in the literature,[42–44] reflecting the challenge faced by healthcare providers in addressing the nuanced needs of people living with Long COVID within the constraints of limited appointment times. Such negative interactions can significantly impact mental health, impeding the overall well-being and recovery of those navigating the complexities of their condition.[45]

Similar issues in patient–provider interactions within the context of Long COVID care have been described in other studies globally, emphasising the pressures on healthcare systems exacerbated by the pandemic.[44 46 47] The existing backlog of waiting lists, compounded by the strains of the COVID-19 pandemic, has strained healthcare resources, possibly contributing to the reported challenges in addressing Long COVID comprehensively within primary care settings. However, it is important to consider whether an individual's expectations align with the practical capacities of clinicians. Nevertheless, overall, these dynamics emphasise the urgent need for tailored approaches to address Long COVID within healthcare systems and fostering improved communication between both parties to align expectations with achievable outcomes.

Our study found that the establishment and use of Long COVID-specific online groups (both patient led and provider led) emerge as a noteworthy peer-support mechanism, facilitating connection and information-sharing among people living with Long COVID. These findings resonate with current literature, highlighting the value of online platforms in fostering a sense of community and understanding among people with Long COVID. The positive impact observed in our study aligns with the observations from other studies[48 49] where they have emphasised the therapeutic benefits of peer support networks in Long COVID management, acknowledging their role in reducing feelings of isolation and offering practical insights. The potential for these digital spaces to serve as outlets for shared experiences and coping strategies reflects a promising avenue for people living with Long COVID navigating the complexities of this condition, affirming the significance of such platforms within the broader landscape of patient-centred care. Nevertheless, while these online groups can be an avenue for peer support, given their potential for a source of conflicting information and misinformation, there remains a demand for accurate, up-to-date information to ensure individuals have reliable and current data which could support medical management.[50 51]

Lastly, a lack of clinical understanding of Long COVID among healthcare providers, as reported (or experienced) by people with Long COVID, aligns with previous findings.[52] This underscores the persistent challenge in understanding Long COVID, signifying the need for continued research and standardised approaches to improve patient care and outcomes in this complex condition.

The third theme aligns with discussions in the literature surrounding specialised post-COVID-19 services. This theme was divided into two perspectives: that of people with Long COVID and healthcare professionals providing care within the post-COVID-19 service.

In our interviews, we found that diverse models of post-COVID-19 services were being used, where some operate primarily face to face (post-COVID-19 services a and c) and others adopt hybrid approaches (post-COVID-19 services b, d, e and f), suggesting flexibility to tailor care to individual needs. The hybrid nature of post-COVID-19 services potentially reflects responsiveness to patient requirements, ranging from entirely virtual to incorporating in-person components such as home visits when necessary. Additionally, considerations such as pandemic-related infection control measures and staffing constraints within these post-COVID-19 service models may have influenced the preference and practicality of a virtual service for certain sites and individuals.[53]

For people with Long COVID, the positive perception of expert care within these clinics echoes the sentiments expressed by recent research[54] stressing the importance of multidisciplinary care teams in addressing the complex and varied symptomatology of Long COVID. While immediate clinical improvement may not always be observed, many people living with Long COVID described the transformative nature of their interactions with healthcare professionals in these specialised clinics. Importantly, for many, it was a place where they were given time to discuss their symptoms and where they were validated and understood. This finding agrees with existing literature,[55 56] reinforcing the crucial role of empathy in health and social care professionals. We underscore its inherent importance in any service, with a particular emphasis on its heightened relevance in situations marked by clinical uncertainty. Further, our findings resonate with existing literature that reflects the pivotal role that post-COVID-19 services play in addressing the complex needs of people living with Long COVID. Our study's insights also align with recommendations from previous studies[57 58] where an emphasis on the importance of multidisciplinary care teams and specialised expertise in tackling the intricate symptomatology of Long COVID was given.

For the virtual post-COVID-19 services, drawbacks potentially include limited medical examination, challenges in effective communication and the risk of exacerbating health inequalities due to disparities in technology access.[59] The lack of hands-on assessments in virtual care may lead to delays in diagnosis or treatment and the digital nature of interactions may limit emotional connections.

Additionally, the digital divide could exclude individuals with limited access to technology, hindering inclusivity in these support networks.[60] Addressing these concerns is crucial for ensuring a comprehensive and equitable approach to managing the complexities of Long COVID.

Healthcare professionals working within the post-COVID-19 service reported clinical uncertainty, 'learning on the job' and a lack of resources—issues all previously noted in other studies.[52 61] Nevertheless, given the largely positive reviews of the post-COVID-19 services, there was a sense that the healthcare professionals in the post-COVID-19 service were able to align their clinical approach with the diverse expectations of those under their care. Striking a balance between patient expectations and the intricacies of their conditions has been well reported in the literature—an essential approach for reducing patient dissatisfaction and potentially improving health outcomes.[52 62] Although there remains clinical uncertainty with respect to Long COVID, efforts to address best practices towards assessing, treating and rehabilitating have been made with a training programme developed by NHSE, which launched in June 2023.[63]

We agree with suggestions that as the body of Long COVID evidence and diagnostic criteria evolves, there may be a need to revise referral criteria for specialist services to ensure the most efficient allocation of resources.[52 64] Similarly, self-referral to post-COVID-19 clinics could be effective if these clinics could adequately manage the demand, which hinges on factors such as staffing and other available resources. However, the viability of such an approach is likely to vary since it heavily relies on the specific capabilities of each post-COVID-19 service and may not be universally applicable.

## Limitations

We acknowledge potential limitations in our methodology. Voluntary participation and its potential to introduce self-selection bias (where individuals who participated differ in relevant clinical characteristics from those who do not) is a limitation.[65] Additionally, the focus on English-speaking participants may exclude those who primarily communicate in other languages, potentially overlooking unique perspectives within linguistically diverse communities. As this study recruited people attending post-COVID-19 services only, this excludes opinions and experiences of those who could not access these services, but were/are living with Long COVID. This limitation could result in a lack of representativeness and under-representation of certain subgroups. Nevertheless, our sample's demographics appear to be aligned with the utilisation of post-COVID-19 services by age, gender, ethnicity in England, as recorded by NHSE.[66]

However, the under-representation of males is a potential limitation, which might limit the comprehensiveness of our findings and hinder a complete understanding of gender-specific experiences. The over-representation of white British/white ethnicities potentially limits the generalisability of our findings to more diverse populations, particularly as evidence suggests that ethnic minorities have been disproportionately impacted by COVID-19[67] and Long COVID has been reported across various ethnic groups.[68] The high representation of individuals with higher education levels and healthcare employment introduces a potential bias, as experiences and access to healthcare resources may differ for those with varied educational backgrounds and occupational statuses.[69] Further, the limited age range of participants may not fully capture the experiences of younger or older individuals with Long COVID.

Overall, although this population is not necessarily representative of the populations who have developed Long COVID at high rates,[1] these findings offer a particular lens through which to understand Long COVID, providing valuable insights into the experiences of this specific subgroup.

## Conclusions

Our study reveals examples of both good care practices and challenges within healthcare provision for people with Long COVID. We underscore the urgent need for a patient-centred approach to Long COVID, addressing its diverse impacts on physical, mental health, social and environmental factors. Effective communication, specialised expertise and comprehensive support systems are pivotal—elements requiring integration into tailored interventions and policies that can holistically address the needs of people with Long COVID. Ensuring continued, sustainable access to specialised care for people with Long COVID is needed to address their multifaceted needs. The success of post-COVID-19 services hinges on the ability to offer ongoing rehabilitation that extends beyond immediate recovery, acknowledging the long-term impacts and complexities associated with this condition.

**Acknowledgements** Our sincere thanks to all participants in this study for their time and the information they provided. This study is part of the STIMULATE-ICP study. The STIMULATE-ICP consortium: Amitava Banerjee, Elizabeth Murray, Hakim-Moulay Dehbi, Hugh Montgomery, Sarah Clegg, Henry Goodfellow, Mel Ramasawmy, Yi Mu, Sampath Weerakkody, Ileana Selejan, David Sunkersing, Ashkan Dashtban, Paula Lorgelly, Melissa Heightman, Toby Hillman, Emma Wall, Caroline Leigh-Watkins, Denise Forshaw, Gordon Prescott, Gail Allsopp, Mark Gabbay, Gregory Lip, Dan Cuthbertson, Dan Wootton, Nefyn Williams, Mike G Crooks, Angela Green, Christina van der Feltz-Cornelis, Jenny Sweetman, Han-I Wang, Natalie Smith, Kamlesh Khunti, Lauren O'Mahoney, Rachael Evans, William D Strain, Rachel Botell, Nisreen Alwan, Donna Clutterbuck, Marija Pantelic, Chris Robson, Mike Brady, Rajarshi Banerjee, Cat Kelly, Angela Barone, Johannes Alberts, Rob Suriano, Lyth Hishmeh, Emily Attree, Jasmine Hayer, Rita Mallinson Cookson, Rachel Hext, Andrew Williams, Rachel Williams, Mag Leahy, Antony Loveless, Clare Loveless, Kim Horstmanshof, Ewen Brennan, Amanpreet Sarna, Manuel Gomes, Andrew Clegg, Valerio Benedetto, Melissa Dalton, Fauzia Begum, Fidan Turk, Michele Pansini, Anu Chandra, Gemma Clunie, Dominic Crocombe, Shane McAuliffe, Michelle Quaye, Farhat Gilani, Yusuf Jaami and Nikki Cleary. An up-to-date version of Consortium members can be found on https://www.stimulate-icp.org/team. STIMULATE-ICP can be contacted at: info@stimulate-icp.org.

**Contributors** AB and MH and the STIMULATE-ICP consortium conceived and designed the study. DS and MR conducted semistructured interviews with both people with Long COVID and healthcare professionals. DS conducted data analysis and drafted the paper. MR, NAA, DC, YM, KH, AB and MH provided constructive feedback on drafts of the manuscript and approved the final version. DS is the guarantor.

**Funding** This research was funded by NIHR (COV-LT2-0043) as part of the STIMULATE-ICP study.

**Disclaimer** The views expressed in this publication are those of the author(s) and not necessarily those of the National Institute for Health and Social Care Research or the Department for Health and Social Care. The funders had no role in the design of the study; in the collection, analyses or interpretation of data; in the writing of the manuscript; or in the decision to publish the results.

**Competing interests** AB has received research funding from NIHR, British Medical Association, Astra Zeneca, National Institute of Aging and European Union. AB is a Trustee of Long COVID SoS. NAA has contributed in an advisory capacity to the WHO and the European Union Commission's Expert Panel on effective ways of investing in health meetings in relation to post-COVID-19 condition. MH is National Specialty Advisor for Long Covid, NHS England. All other authors report no competing interests.

**Patient and public involvement** Patients and/or the public were involved in the design, or conduct, or reporting, or dissemination plans of this research. Refer to the Methods section for further details.

**Patient consent for publication** Not applicable.

**Ethics approval** This study involves human participants and ethical approval (IRAS Project ID: 303958) was granted by the Health Research Authority (HRA) and South Central—Berkshire Research Ethics Committee (REC Reference: 22/SC/0047). Participants gave informed consent to participate in the study before taking part.

**Provenance and peer review** Not commissioned; externally peer reviewed.

**Data availability statement** Data are available on reasonable request.

**ORCID iDs**
David Sunkersing http://orcid.org/0000-0001-9010-1435
Amitava Banerjee http://orcid.org/0000-0001-8741-3411

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
