## [Reviewer comments · BMJ Open]

ARTICLE DETAILS

TITLE (PROVISIONAL)	What is current care for people with Long COVID in England? A qualitative interview study.
AUTHORS	Sunkersing, David; Ramasawmy, Mel; Alwan, Nisreen; Clutterbuck, Donna; Mu, Yi; Horstmanshof, Kim; Banerjee, Amitava; Heightman, Melissa

VERSION 1 – REVIEW

REVIEWER	Baz, Sarah University of York, Health Sciences
REVIEW RETURNED	27-Nov-2023

GENERAL COMMENTS	Review comments Abstract The abstract clearly summarises the study and main points. Participants – given that healthcare professionals are recruited from 6 sites the number of healthcare professionals interviewed is quite low, I would have expected more, at least 20. Could you please justify why only 15 took part – were they small teams for example? This can be expanded upon in the methods section. Strengths and limitations of this study You state a key strength is that this is the first qualitative study that has perspectives from both people with Long Covid and healthcare professionals, however there is another published studies who have already done this – see Baz et al (2022) titled: 'I don't know what to do or where to go'. Experiences of accessing healthcare support from the perspectives of people living with Long Covid and healthcare professionals: A qualitative study in Bradford, UK. Instead, could this potentially be the first study that interviews healthcare professionals from 6 specialist LC services, looks at more than one city. Introduction A great introduction clearly informing the reader of what Long Covid is and providing context to the research. Methods I would repeat in the methods section the number of people you have interviewed. In the methods section there is more detail on people with Long Covid, could you also tell the reader more about recruiting healthcare professionals – for example, how many were approach and any reasons for not taking part. How many from each site? Results Really interesting way to incorporate the quotes. The results section is very interesting and considers the multi-dimensional impacts of Long Covid and experiences of services. It nicely integrates findings and key narratives from both datasets.
---

	Importance of validation seems to be a key theme which can be brought out more. Discussion You mention the varied trajectories, could you tell us more about this. How do the trajectories vary in your data set? Great to see you have mentioned the structural issues that impact experiences of people with Long Covid, particularly in primary care. This further emphasises the important need of specialist Long Covid clinics. But I was wondering does the barriers in access and not being listened to in primary care settings impact whether people can access the specialist clinics, as I believe GP referral is required, and it's also important for the GP to see them first just in case there are other red flags that are not Long Covid. What implications does this have and what can be recommended so people are getting the specialistic Long Covid services care? Better communication is a good suggestion you've included. Should there be recommendations for self-referral, GPs within clinics where there is less pressure or something else? Was this discussed in interviews with HCPs? Also, as your study recruited people attending post-COVID services this excludes the opinion and experiences of those who could not access it – this could be noted as a limitation of the study.
--	--

REVIEWER	Fowler-Davis, Sally Anglia Ruskin University, Health, Medicine and Social Care
REVIEW RETURNED	03-Jan-2024

GENERAL COMMENTS	Thanks for the opportunity to review this paper. It is clearly attempting to present the concerns of patients and staff from the perspective of Long Covid services but I felt that the analysis, presentation of results and discussion were extremely limited and had concerns that what might have been presented as a critical issue associated with the long-term care planning for patient care had been missed. Abstract- the information about the qualitative analysis should be added and the conclusion 'toned' down. I would argue that this paper does not provide a 'roadmap' and does not allow more targeted interventions. Introduction- Please use further references reflecting the patient role in identifying Long Covid and literature associated with narrative and supportive peer interactions that have been a major and important part of understanding the diagnosis. This is not a UK study, with inclusion of only English settings and does not sufficiently provide the historical context of rapid service re-design to meet the need coming out of the pandemic Methods- the description of recruitment is inadequate and fails to inform the reader of the limitations . Methodology to underpin the choice of semi-structured interviews is needed and an appropriate sampling strategy offered, that reflects the question. The methods are reliant on nomination and self selection and as a consequence there is a very narrow understanding presented here- not just the absence of those who are young and old! Ethics is noted and interview schedule is provided. Analysis - Can you provide a reference for "An inductive, data-driven approach" and specify how the PPI was involved- where
---

	these people with Long Covid? The lack of information about why the staff and patient data was analysed together is the most perplexing element of the analysis- why was the study analysed in this way and how is this justified? Results - Sampling- You had 21 participants but why so few from 6 services? 21 relative to how many possible participants- how do you evaluate your sample? your sample group of patients were defined by highly educated white and many health professionals and yet this dominant group is not discussed or presented as offering a particular perspective. See other papers reflecting on the variation in treatment when working with patients who are poor, on zero hours contracts and with greater ethnic diversity. It is very unclear how the presentation of themes was developed and many of the quotes did not seem to illustrate the theme or the sub-themes. I suggest complete revision to the presentation of findings, with more carefully and illustrative selection of the quote and much shorter examples. The titles of the themes seemed strange and again, the combination of staff and patient responses together seemed ignorant of the very different perspectives on the service delivery. For example, in Theme 3 I could not see how avoidance was linked to GP activity and overall the commentary appeared to reflect a shared frustration with the care offered and received. Discussion - I would strongly recommend the separation of service user and service provider perspectives and further discussion about the importance of the peer to peer support offer. The uncertainty created by a lack of definitive treatment evidence and the highly individual and sometimes traumatic nature of the disability caused is not poorly reflected in the discussion. The expectations of patients and the way that staff manage and support/care for patients, given the complexity of the challenge seems to be under-reported and the conclusion inadequately represents the critical challenge of sustaining services and offering rehabilitative support. Smaller points include the use of terminology, "Covid sufferer" P14 line 19 seems inappropriate but perhaps reflects a somewhat limited inductive understanding of the data and the means that can be derived from it. The conclusion needs to be thoroughly revised to reflect the implications of the study for patients and staff, it is insufficient to say that there is more research needed.
--	---

VERSION 1 – AUTHOR RESPONSE

Reviewer: 1 Dr. Sarah Baz, University of York Comments to the Author: Review comments	Response
---	------------------------

Abstract The abstract clearly summarises the study and main points. Participants – given that healthcare professionals are recruited from 6 sites the number of healthcare professionals interviewed is quite low, I would have expected more, at least 20. Could you please justify why only 15 took part – were they small teams for example? This can be expanded upon in the methods section.	Comments noted. Recruitment sample size expanded upon in the methods section.
Strengths and limitations of this study You state a key strength is that this is the first qualitative study that has perspectives from both people with Long Covid and healthcare professionals, however there is another published studies who have already done this – see Baz et al (2022) titled: 'I don't know what to do or where to go'. Experiences of accessing healthcare support from the perspectives of people living with Long Covid and healthcare professionals: A qualitative study in Bradford, UK. Instead, could this potentially be the first study that interviews healthcare professionals from 6 specialist LC services, looks at more than one city.	This key strength (first bullet point in 'Strengths and Limitations of This Study') has been removed in accordance with the Editor's suggestion.
Introduction A great introduction clearly informing the reader of what Long Covid is and providing context to the research.	Although Reviewer 1 has suggested no changes, we have refined the introduction for greater clarity and have addressed the comments from Reviewer 2.
Methods I would repeat in the methods section the number of people you have interviewed. In the methods section there is more detail on people with Long Covid, could you also tell the reader more about recruiting healthcare professionals – for example, how many were approached and any reasons for not taking part. How many from each site?	Subheadings have now been added for greater clarity. Additional information has been added re. 'Recruitment of Healthcare Professionals' as well as details of lower anticipated sample size. We have now given further details of the number of people we have interviewed (in addition to how many per site and reasons for not participating), although we believe this is better placed in the results section.
Results Really interesting way to incorporate the quotes. The results section is very interesting and considers the multi-dimensional impacts of Long Covid and experiences of services. It nicely integrates findings and key narratives	Additional sentences have been added re. 'validation' theme. We have also included a generic patient pathway that was used by the post-COVID sites (Figure 1).

from both datasets. Importance of validation seems to be a key theme which can be brought out more.	
Discussion You mention the varied trajectories, could you tell us more about this. How do the trajectories vary in your data set? Great to see you have mentioned the structural issues that impact experiences of people with Long Covid, particularly in primary care. This further emphasises the important need of specialist Long Covid clinics. But I was wondering does the barriers in access and not being listened to in primary care settings impact whether people can access the specialist clinics, as I believe GP referral is required, and it's also important for the GP to see them first just in case there are other red flags that are not Long Covid. What implications does this have and what can be recommended so people are getting the specialistic Long Covid services care? Better communication is a good suggestion you've included. Should there be recommendations for self-referral, GPs within clinics where there is less pressure or something else? Was this discussed in interviews with HCPs? Also, as your study recruited people attending post-COVID services this excludes the opinion and experiences of those who could not access it – this could be noted as a limitation of the study.	'Trajectories' was only meant to refer to that reported in the literature. 'Varied trajectory' has been removed for clarity, as this was not formally analysed in our study. Suggestion of self-referral has been added to the discussion, though with the acknowledgement that this will be dependent on individual post-COVID services. Suggested limitation (i.e. those who could not access post-COVID services) has been added). We have additionally now included discussion of the models of post-COVID services (i.e. face-to-face/hybrid).

Reviewer: 2 Dr. Sally Fowler-Davis, Anglia Ruskin University	Response
Comments to the Author: Thanks for the opportunity to review this paper. It is clearly attempting to present the concerns of patients and staff from the perspective of Long Covid services	Thank you for your comments and suggestions. We have taken on board all reviewer comments and revised the paper accordingly.

but I felt that the analysis, presentation of results and discussion were extremely limited and had concerns that what might have been presented as a critical issue associated with the long-term care planning for patient care had been missed.	
Abstract- the information about the qualitative analysis should be added and the conclusion 'toned' down. I would argue that this paper does not provide a 'roadmap' and does not allow more targeted interventions.	'Data analysed using thematic analysis' added to 'Design' part of Abstract. Conclusion has been revised/toned down. Now better reflects the implications for patients and staff.
Introduction- Please use further references reflecting the patient role in identifying Long Covid and literature associated with narrative and supportive peer interactions that have been a major and important part of understanding the diagnosis. This is not a UK study, with inclusion of only English settings and does not sufficiently provide the historical context of rapid service re-design to meet the need coming out of the pandemic	Introduction now improved by adding references (as suggested) reflecting the patient role in identifying Long COVID/supportive peer interactions. Reference to this study taking place in the UK corrected to 'England'. The introduction now includes a more detailed historical context of the rapid service re-design regarding Long COVID care, as suggested. Overall, the introduction has been refined for clarity.
Methods- the description of recruitment is inadequate and fails to inform the reader of the limitations . Methodology to underpin the choice of semi-structured interviews is needed and	Details of recruitment have now been expanded (with separate sections for people with Long COVID and Healthcare professionals). Further details on why semi-structured interviews were chosen for this study has been added. Addition of voluntary response sampling as the sampling strategy added, with the rationale explained (for both Long COVID patients and

an appropriate sampling strategy offered, that reflects the question. The methods are reliant on nomination and self selection and as a consequence there is a very narrow understanding presented here- not just the absence of those who are young and old! Ethics is noted and interview schedule is provided.	healthcare professionals). Details and acknowledgement of limitations of this sampling strategy has also been added.
Analysis - Can you provide a reference for "An inductive, data-driven approach" and specify how the PPI was involved- where these people with Long Covid? The lack of information about why the staff and patient data was analysed together is the most perplexing element of the analysis- why was the study analysed in this way and how is this justified?	Reference to 'inductive, data-driven approach' now included. Details of PPI group involvement has been detailed. Further information on their background (i.e. whether they have had Long COVID or not) and specific reference to this study has now been included. The data analysis section has been refined for clarity. The approach of generating shared overarching themes for both people with Long COVID and healthcare professional perspectives was done to enable a deeper understanding of shared and divergent perspectives – an approach used in other studies https://onlinelibrary.wiley.com/doi/10.1111/jpm.12497 and https://www.sciencedirect.com/science/article/pii/S0163834313000170). This has been added to this section.

Results - Sampling- You had 21 participants but why so few from 6 services? 21 relative to how many possible participants- how do you evaluate your sample? your sample group of patients were defined by highly educated white and many health professionals and yet this dominant group is not discussed or presented as a offering a particular perspective. See other papers reflecting on the variation in treatment when working with patients who are poor, on zero hours contracts and with greater ethnic diversity.

Is is very unclear how the presentation of themes was developed and may of the quotes did not seem to illustrate the theme or the sub-themes. I suggest complete revision to the presentation of findings, with more carefully and illustrative selection of the quote and much shorter examples. The titles of the themes seemed strange and again, the combination of staff and patient responses together seemed ignorant of the very different perspectives on the service delivery. For example, in Theme 3 I could not see how avoidance was linked to GP activity and overall the commentary appeared to reflect a shared frustration with

Re. sample size – The principle of saturation guided the sample size, and semi-structured interviews were conducted until additional participants did not yield new information. This has been added under ‘Recruitment of people with Long COVID’. Further details have also been added under ‘Recruitment of Healthcare Professionals’.

We agree that the sample is predominantly white, educated and with a healthcare professional background, which is a particular perspective. These limitations have now been discussed in the ‘Limitations’ section.

As suggested, the presentation of findings has now been completely revised to be much clearer, as detailed below:

Quotes have been shortened, where appropriate. Some quotes have been replaced with ones that reflect the subtheme better (e.g., ‘Mental Health’ in Theme 1).

Some of the subtheme names have now been revised (Theme 2 Subtheme ‘Avoidance’ changed to ‘Frustrations With Primary Care Service’ to better reflect findings’; Theme 2 Subtheme ‘Inefficiencies’ changed to ‘Waiting Times’; Theme 2 Subtheme ‘Communication’ changed to ‘Negative Clinical Engagements’; Theme 2 Subtheme ‘Clinical Expertise’ changed to ‘Positive Clinical Engagements’; Theme 2 ‘Challenges changed to ‘Clinical Challenges’.

Theme 3 Subthemes are now split into two defined categories: Post-COVID Service Experiences and Post-COVID Experiences (HCP) Perspective.

Theme names have been reworded to have more focus. Theme 1: Understanding Long COVID: Symptoms and Lived Realities; Theme 2: Long COVID Care: Non-Specialist Service Experiences; Theme 3: Long COVID Care: Post-COVID Service Experiences.

In addition to this, only Themes 1 and 3 include perspectives from healthcare professionals, for greater clarity. These perspectives have been separated in the table from the perspectives of people with Long COVID.

the care offered and received.	
Discussion - I would strongly recommend the separation of service user and service provider perspectives and further discussion about the importance of the peer to peer support offer. The uncertainty created by a lack of definitive treatment evidence and the highly individual and sometimes traumatic nature of the disability caused is not poorly reflected in the discussion. The expectations of patients and the way that staff manage and support/care for patients, given the complexity of the challenge seems to be under-reported and the conclusion inadequately represents the critical challenge of sustaining services and offering rehabilitative support.	The discussion has now been revised and separates perspectives from people with Long COVID and healthcare professionals. The importance of peer to peer support has been discussed within theme 2. The mental health impact has been further highlighted in the theme 1 discussion. We have also added reference to the uncertainty created by a lack of an effective treatment in this discussion. Throughout the discussion, we refer to the 'personalised care' which reflects the individual nature of the condition. 'Patient' expectations versus the way healthcare professionals manage/support/care has been further mentioned (both within theme 2 and 3 discussions) The conclusion has now been revised to recommend continued and sustainable access to post-COVID clinics and ongoing rehabilitation.

Smaller points include the use of terminology, "Covid sufferer" P14 line 19 seems inappropriate but perhaps reflects a somewhat limited inductive understanding of the data and the means that can be derived from it. The conclusion needs to be thoroughly revised to reflect the implications of the study for patients and staff, it is insufficient to say that there is more research needed.	Although the terminology 'Long COVID sufferer' is a commonly used term in online reports and in published literature, we accept that this could be phrased better. The two instances of this have been changed to 'people with Long COVID'. Re. conclusion – we understand that this refers to the conclusion in the abstract. Conclusion has been revised to focus on implications for patients and staff.
--	---

VERSION 2 – REVIEW

REVIEWER	Fowler-Davis, Sally Anglia Ruskin University, Health, Medicine and Social Care
REVIEW RETURNED	17-Mar-2024
GENERAL COMMENTS	This paper has been improved since the first review. Please re-write the conclusion and express the findings in a more academic way with additional explanation of limitations of the methods more fully

VERSION 2 – AUTHOR RESPONSE

Reviewer: 2 Dr. Sally Fowler-Davis, Anglia Ruskin University	Response
Comments to the Author: This paper has been improved since the first review. Please re-write the conclusion and express the findings in a more academic way with additional explanation of limitations of the methods more fully	Thank you for your comments and suggestions. As suggested, we have re-written the conclusion, expressing the findings more academically. The limitations section has been revised and expanded to highlight the limitations of the methodology more fully.

VERSION 3 – REVIEW

REVIEWER	Fowler-Davis, Sally Anglia Ruskin University, Health, Medicine and Social Care
REVIEW RETURNED	29-Apr-2024
GENERAL COMMENTS	This paper has been substantially improved